# Agreement between two large pan-cancer CRISPR-Cas9 gene dependency data sets

Joshua M. Dempster [1], Clare Pacini [2,3], Sasha Pantel[1], Fiona M. Behan [2,3], Thomas Green[1], John Krill-Burger [1], Charlotte M. Beaver[2], Scott T. Younger[1], Victor Zhivich[1], Hanna Najgebauer[2,3], Felicity Allen[2], Emanuel Gonçalves[2], Rebecca Shepherd[2], John G. Doench [1], Kosuke Yusa[2,7], Francisca Vazquez [1], Leopold Parts[2,4], Jesse S. Boehm [1], Todd R. Golub [1,5], William C. Hahn [1,5], David E. Root [1], Mathew J. Garnett[2,3], Aviad Tsherniak [1]* & Francesco Iorio [2,3,6]*

Genome-scale CRISPR-Cas9 viability screens performed in cancer cell lines provide a systematic approach to identify cancer dependencies and new therapeutic targets. As multiple large-scale screens become available, a formal assessment of the reproducibility of these experiments becomes necessary. We analyze data from recently published pan-cancer CRISPR-Cas9 screens performed at the Broad and Sanger Institutes. Despite significant differences in experimental protocols and reagents, we find that the screen results are highly concordant across multiple metrics with both common and specific dependencies jointly identified across the two studies. Furthermore, robust biomarkers of gene dependency found in one data set are recovered in the other. Through further analysis and replication experiments at each institute, we show that batch effects are driven principally by two key experimental parameters: the reagent library and the assay length. These results indicate that the Broad and Sanger CRISPR-Cas9 viability screens yield robust and reproducible findings.

[1] Broad Institute of MIT and Harvard, Cambridge, MA 02142, USA. [2] Wellcome Sanger Institute, Wellcome Genome Campus, Hinxton, Cambridge CB10 1SA, UK. [3] Open Targets, Wellcome Genome Campus, Hinxton, Cambridge CB10 1SA, UK. [4] Department of Computer Science, University of Tartu, 50090 Tartu, Estonia. [5] Dana-Farber Cancer Institute, Boston, MA 02215, USA. [6] Human Technopole, 20157 Milano, Italy. [7] Present address: Stem Cell Genetics, Institute for Frontier Life and Medical Sciences, Kyoto University, Kyoto 606-8507, Japan. *email: aviad@broadinstitute.org; fi1@sanger.ac.uk

Despite recent advances in cancer research, most cancer patients still have no clinical indications for approved targeted therapies[1]. Expanding precision oncology to the general patient population will require identifying and exploiting many new genomic targets. To tackle this problem, large-scale pharmacogenomic screenings have been performed across panels of human cancer cell lines[2,3]. The advent of genome editing by CRISPR-Cas9 technology has allowed extending these studies beyond currently druggable targets with precision and scale[4,5]. Pooled CRISPR-Cas9 screens employing genome-scale libraries of single-guide RNAs (sgRNAs) are being performed on growing numbers of cancer in vitro models[6–12]. The output of these screens can be used to identify and prioritize new cancer therapeutic targets[13]. However, fully characterizing genetic vulnerabilities in cancers is estimated to require thousands of genome-scale screens[14].

We present a comparative analysis of data sets derived from the two largest independent CRISPR-Cas9 based gene-dependency screening studies in cancer cell lines published to date[13,15,16], part of the Cancer Dependency Map effort[17,18]. The analysis aims to assess the concordance of these data sets and that of the analytical outcomes they yield when investigated individually. To this aim, our computational strategy includes comparisons at different levels of data-processing and abstraction: from gene-level dependencies to molecular markers of dependencies, and genome-scale cell line profiles of dependencies. Lastly, we shed light on the differences in the experimental settings that give rise to batch effects across independent studies of this kind, discerning between biological and technical confounding factors.

## Results

**Overview of data sets and comparison strategy.** We compared two sets of pooled genome-scale CRISPR-Cas9 drop out screens in cancer cell lines, generated at the Broad Institute and the Sanger Institute through independently designed experimental pipelines (detailed in Fig. 1a, Supplementary Data 1 and Supplementary Methods), considering 147 cell lines and 16,733 genes screened independently by both institutes (Supplementary Data 2). We performed comparisons of individual gene scores, quantifying the reduction of cell viability upon gene inactivation via CRISPR-Cas9 targeting; of profiles of such scores across cell lines (gene dependency profiles); of profiles of such scores across genes in individual cell lines (cell line dependency profiles).

We calculated gene scores using three different strategies. First, we considered fully processed gene scores, available for download from the Broad[17] and Sanger[13,18] Cancer Dependency Map webportals (processed data). Because data processing pipelines vary significantly between the two data sets, we also examined minimally processed gene scores, generated by computing median sgRNA abundance fold changes for each gene (unprocessed data). Lastly, we applied the established batch correction method ComBat[19] to the unprocessed gene scores to remove experimental batch effects between the data sets. This is achieved by ComBat through aligning gene means and variances between the data sets using an empirical Bayes framework. We refer to this form of the data as the batch-corrected gene scores.

**Agreement of gene scores.** We found concordant gene scores across all genes and cell lines with Pearson correlation = 0.658, 0.627, and 0.765, respectively for processed, unprocessed and batch-corrected data ($p$-values below machine precision in all cases, $N = 2,465,631$, Fig. 1b). Spearman correlations across the different comparisons were 0.347, 0.411, and 0.551 respectively, again significant below machine precision. The reproducibility of

gene scores between the two data sets can be considered a function of two variables: the mean dependency across all cell lines for each gene (relevant to infer common dependencies), and the patterns of scores across cell lines for each gene (relevant to predict selective oncology therapeutic targets). Mean gene scores among all cell lines showed excellent agreement (Supplementary Fig. 1a), with Pearson correlation = 0.784 and 0.818, respectively for processed andunprocessed data ($p$ below machine precision in both cases using SciPy's beta distribution test; $N = 16,773$). The effect of ComBat correction on our data is to align gene means and variances (Supplementary Fig. 1b). As expected, after Com-Bat correction the Pearson correlation of gene means was = 0.9997, and the correlation of gene standard deviations (SDs) was = 0.957.

We further tested whether it was possible to recover consistent sets of common dependencies. To this end, we defined as common dependencies those genes that rank among the top dependencies when considering only their 90th percentile of least dependent cell lines, with the score threshold for top dependencies determined by the local minimum in the data (Fig. 1c). For the unprocessed data, the Broad and Sanger jointly identify 1,031 common dependency genes (Supplementary Data 3). 260 putative common dependencies were only identified by the Sanger and 397 were only identified by the Broad (Cohen's kappa = 0.737, Fisher's exact test p-value below machine precision, $N = 16,773$, Fig. 1d).

**Agreement of selective gene score profiles across cell lines.** In both studies, most genes show little variation in their scores across cell lines. Thus we expect low shared variance even if most scores are numerically similar between the data sets[20]. Accordingly, we focused on a group of genes for which the score variance across lines is of potential biological interest. These are genes whose dependency profile suggests a strong biological selectivity in at least one of the two unprocessed data sets, identified using the Likelihood Ratio Test (NormLRT) test introduced in McDonald et al.[21]. We call these 49 genes Strongly Selective Dependencies (SSDs) (Supplementary Data 4). We evaluated the agreement between gene score patterns across cell lines using Pearson's correlations to test the reproducibility of selective viability phenotypes. Figure 2a illustrates the score patterns for the example cancer genes *MDM4* ($R = 0.820$, beta test $p = 6.91 \times 10^{-37}$), *KRAS* ($R = 0.765$, $p = 1.66 \times 10^{-29}$), *CTNNB1* ($R = 0.803$, $p = 1.92 \times 10^{-34}$), and *SMARCA4* ($R = 0.664$, $p = 4.61 \times 10^{-20}$) with unprocessed data ($N = 147$). For SSDs and unprocessed data, the median correlation was 0.633 and 84% of SSDs showed a correlation greater than 0.4. Five SSDs showed a correlation below 0.2 (*ABHD2*, *CDC62*, *HIF1A*, *HSPA5*, *C17orf64*), and are discussed further below. As expected, correlation across data sets for all genes was lower (median $R = 0.187$, 8.34% genes with $R > 0.4$).

One important use of these screens is to consistently classify cells as dependent or not dependent on selective dependencies. Therefore, we evaluated the agreement of the Broad and Sanger data sets on identifying cell lines that are dependent on each SSD gene. We classified cell lines as dependent on a given gene if its gene score represents a false discovery rate (FDR) less than 0.05 (see the Methods section). Genes scores with greater than 5% FDR are dominated by a large group of scores near zero (Fig. 2c).

The area under the receiver-operator characteristic (AUROC) for recovering binary Sanger dependency on SSDs using Broad gene scores was 0.940 in processed data, 0.963 in unprocessed data, and 0.971 in corrected data; to recover Broad binary dependency from Sanger scores, AUROC scores were 0.918, 0.870, and 0.968 respectively. The recall of Sanger-identified

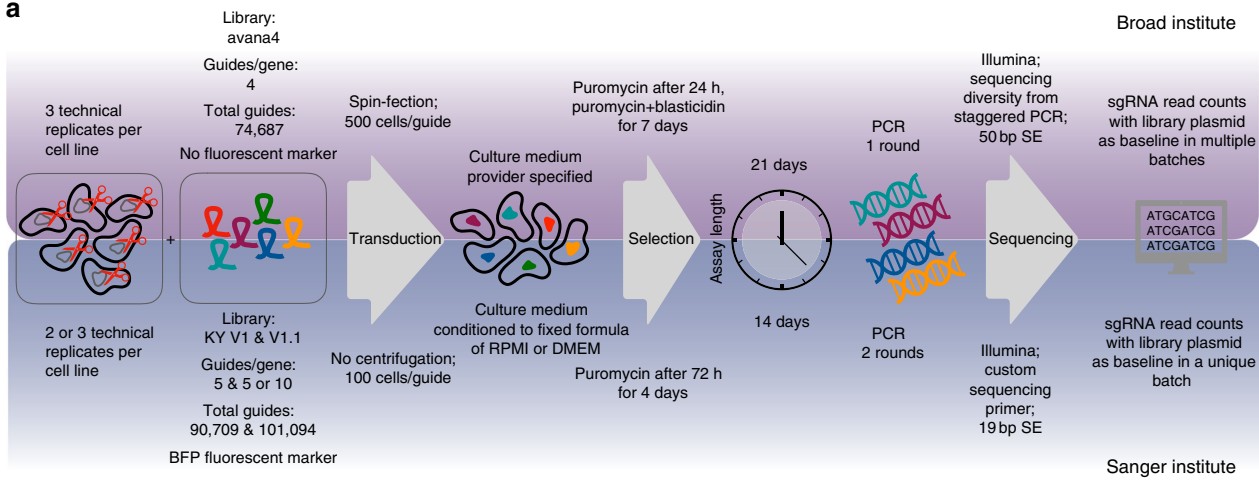

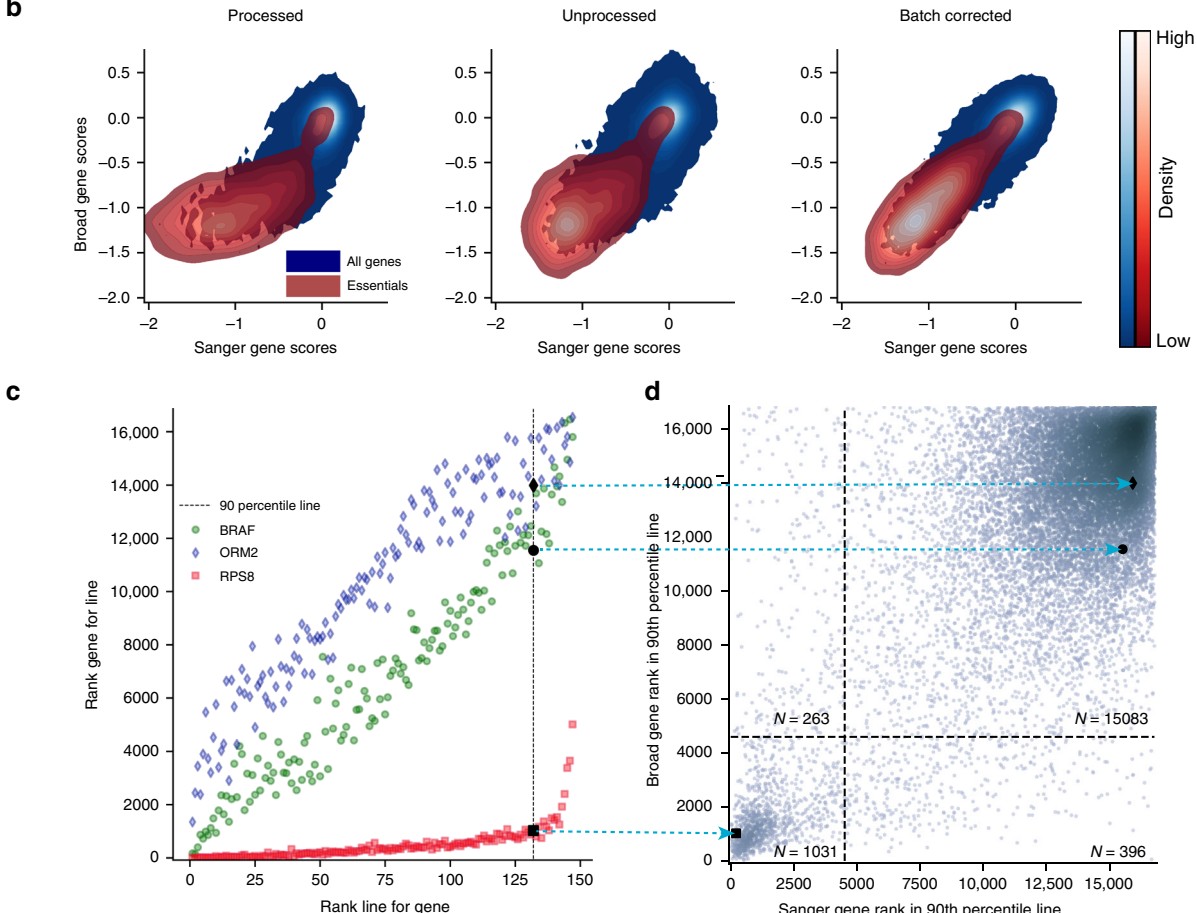

**Fig. 1 Comparison of experimental protocols and gene score results. a** Experimental settings and reagents used in the experimental pipelines underlying the two compared data sets. **b** Densities of individual gene scores in individual cell lines, in the Broad and Sanger data sets, across processing levels. The distributions of gene scores for previously identified essential genes[12] are shown in red. **c** Examples of the relationship between a gene's score rank in a cell line and the cell line's rank for that gene using Broad unprocessed gene scores, with gene ranks in their 90th percentile of least dependent lines highlighted. Cell lines in the 90th percentile of least dependent lines on RPS8 (a common essential gene) still rank this gene among the strongest of their dependencies. **d** Distribution of gene ranks for the 90th percentile of least dependent cell lines for each gene in both data sets. Black dotted lines indicate natural thresholds at the minimum gene density along each axis. The y-axis is equivalent to the y-axis in (**c**) at the 90th percentile mark, as indicated by the arrows.

dependent cell lines in Broad data was 0.781 with precision equal to 0.255 for processed data, 0.775 and 0.258 for unprocessed data, and 0.754 and 0.587 for batch-corrected data (Supplementary Fig. 1c). Agreement is higher than could be expected by chance under all processing regimes (Fisher's exact test $p = 8.99 \times 10^{-43}$

in processed, $9.65 \times 10^{-44}$ in unprocessed, and $5.29 \times 10^{-198}$ in batch-corrected data; $N = 7{,}203$). A large proportion of Broad-exclusive dependent cell lines (53.4% in processed data and 47.7% in unprocessed data) were due to the single gene HSPA5, which is an SSD in Sanger data but a common dependency in Broad data.

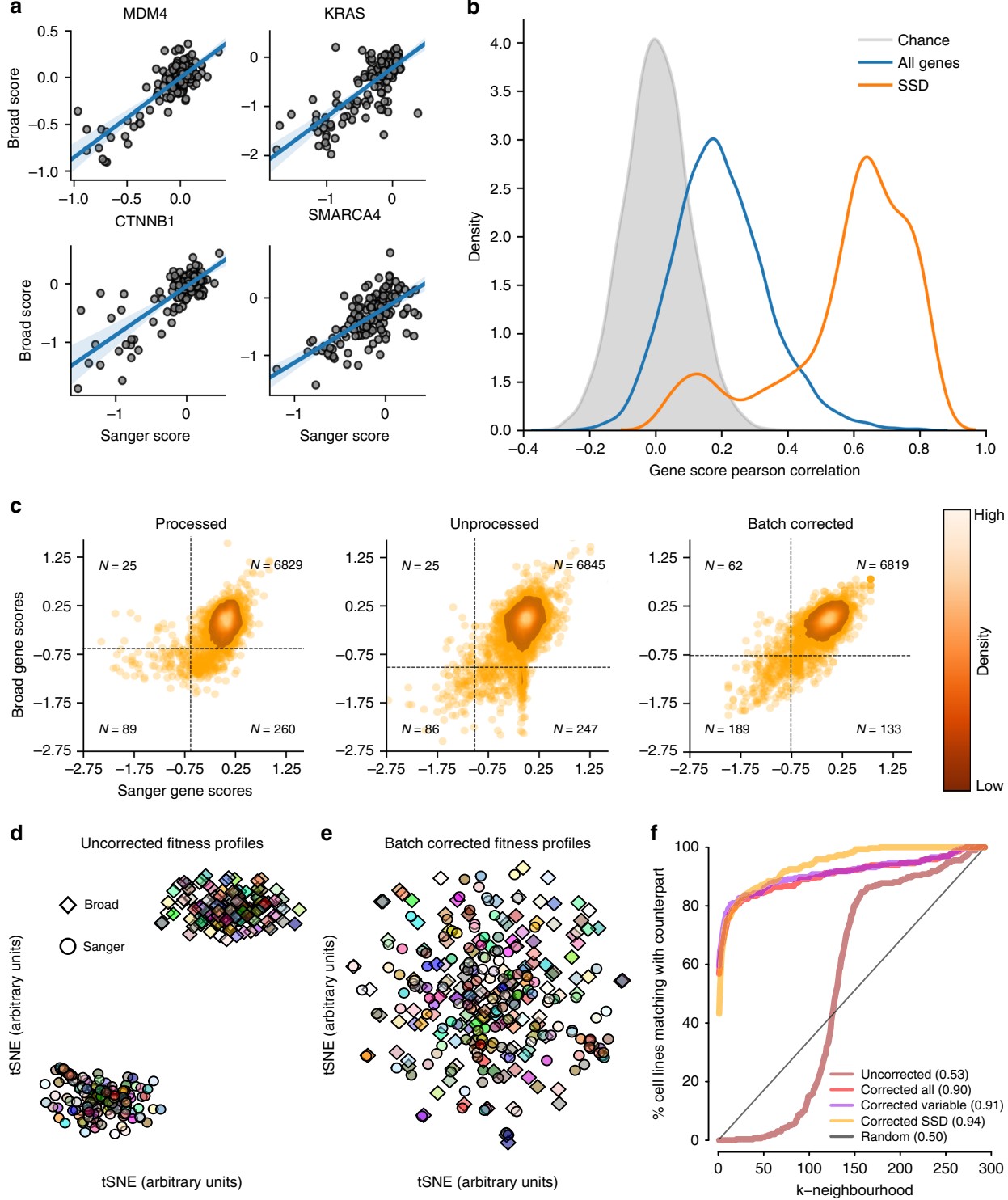

**Fig. 2 Reproducibility of gene and cell line dependency profiles. a** Examples of gene score pattern comparisons for selected known cancer genes.
**b** Distribution of correlations of scores for individual genes in unprocessed data. **c** Gene scores for strongly selective dependencies across all cell lines, with
the threshold for calling a line dependent set at an FDR of 0.05. **d** tSNE visualization of cell lines in unprocessed data based on the correlation between cell
line profiles of gene scores. Colors represent the cell line while shape denotes the study of origin. **e** The same as in (**d**) but for data batch-corrected using
ComBat. **f** Recovery of a cell line's counterpart in the other data set before (Uncorrected) and after correction (Corrected). Value on the y-axis shows
percentages of cell lines whose matching counterpart in the other data set is within its k-nearest cell lines, i.e. the k-neighborhood on the x-axis, based on a
Pearson correlation distance metric. nAUC values are shown in brackets. Three different gene sets were considered to calculate the correlation between
cell lines. First, using all genes (uncorrected and corrected all), second, using genes that are dependencies for at least one cell line (corrected variable) and
third, using strongly selective dependencies (corrected SSD) genes.

Examining SSDs individually, we found median Cohen's kappa for sensitivity to individual SSDs of 0.461 in processed, 0.609 in unprocessed, and 0.758 in batch-corrected data. In unprocessed data, 59.2% of SSDs had Cohen's kappa greater than 0.4, as opposed to 0.03% expected by chance (Supplementary Fig. 1c).

**Agreement of cell line dependency profiles**. Previous literature on reproducibility highlighted the importance of considering agreement along both the perturbation and cell line axes of the data[22–24]. We assembled a combined data set of cell line dependency profiles from both studies and computed all possible pairwise correlation distances between them, using genes that were dependencies in at least one cell line (variable genes). A t-distributed stochastic neighbor embedding (tSNE)[25] visualisation derived from these distance scores is shown in Fig. 2d. For the uncorrected data, we observed a perfect clustering of the dependency profiles by their study of origin, confirming a major batch effect. However, following batch correction, we observed integration between studies and increased proximity of cell lines from one study to their counterparts in the other study (Fig. 2e). To quantify agreement, for each cell line dependency profile in one data set, we ranked all the others (from both data sets) based on their correlation distance to the profile under consideration. For batch-corrected data, 175 of 294 (60%) cell line dependency profiles from one study have their counterpart in the other study as the closest (first) neighbor, and 209 of 294 (71%) of cell lines have it among the five closest neighbors (area under the normalized Recall curve — nAUC — averaged across all profiles = 0.91 for batch-corrected data, and = 0.53 for uncorrected data, Fig. 2f). Similar results were obtained across dependency profiles restricted to different sets of genes, with the best performance obtained when considering SSD genes only (nAUC = 0.94) and worst performances when considering all genes (nUAC = 0.90). The percentage of cell lines matching closest to their counterparts in the other study was 57% when considering all genes and 43% when considering SSD genes. Further, the tSNE plots for each tested gene set showed similar improvement after correction (Supplementary Fig. 2a–b).

The batch correction also aligned numbers of significant (at 5% FDR) dependencies across cell lines between the two data sets (median number of dependencies 2,109 and 1,717 before, and 2,053 and 1,950 after correction, for Broad and Sanger respectively, Supplementary Fig. 3a). The average proportion of dependencies detected in both studies over those detected in at least one study also increased across cell lines from 47.75% to 59.14%. Furthermore, the correlation between cell lines after correction rose above the correlation within each individual screen for each gene set considered (Supplementary Fig. 3b). We finally examined whether the residual disagreement in corrected data might be related to screen quality and if there are tissues for which corresponding cell lines showed a consistently higher/lower agreement across the two studies. We assessed screen quality by computing true positive rates (TPRs) for recovering common essential genes in each cell line with a fixed 5% FDR, determined from the distribution of nonessential genes in the cell line. We found that mean screen quality is a strong predictor of screen agreement for both the uncorrected and batch-corrected data sets (t-test p-values $2.06 \times 10^{-35}$, $4.74 \times 10^{-35}$, N = 147 and adjusted R-squared 0.65, 0.64 for uncorrected and batch-corrected respectively; Supplementary Fig. 3c). In addition, we observed no differences in screen agreement when stratifying cell line based on their tissue of origin (Supplementary Fig. 3d), with screen quality being highly correlated with screen agreement invariantly across tissues (Supplementary Fig. 3e and Supplementary Data 5).

**Agreement of gene dependency biomarkers**. A selective dependency is of limited therapeutic value unless it can be reliably associated with an informative molecular feature of cancer (biomarker). Following a similar approach to that presented by the Cancer Cell Line Encyclopedia and Drug Sensitivity in Cancer consortia[20], we performed a systematic test for molecular-feature/dependency associations on the two data sets. To this aim, we considered a set of Cancer Functional Events consisting of 578 molecular features selected in Iorio et al.[26] based on their clinical relevance and encompassing mutations in high-confidence cancer driver genes, amplifications/deletions of chromosomal segments recurrently altered in cancer, hypermethylated gene promoters, microsatellite instability status, and the tissue of origin of the cell lines (Supplementary Data 5). We considered each of these features in turn and observed its status in the cell lines screened at both Sanger and Broad. Based on this, cell lines were split into two groups (respectively with negative/positive feature) and each of the SSD genes was t-tested for significant differences in gene scores across the obtained two groups of cell lines.

These tests yielded 71 out of 29,350 possible significant associations (FDR < 5%, ΔFC < −1) between molecular features and gene dependency when using the Broad unprocessed data, and 90 when using the Sanger unprocessed data (Supplementary Data 6). Of these, 55 (77% of the Broad associations and 61% of the Sanger ones) were found in both data sets (FET p-value = $9.08 \times 10^{-133}$, Fig. 3a and Supplementary Data 6). The concordance between the associations identified by each study was proportional to the threshold used to define significance (Supplementary Data 7). This was assessed by first considering the associations found significant (FDR < 5%) in one study as positive controls and calculating precision, recall, and sensitivity using a rank predictor based on the p-values obtained in the other study for all associations. We then tested how performance changed when considering increasingly stringent subsets of significant associations as positive controls and found that the most significant associations in one study were the most likely to be recovered in the other (Fig. 3b). Further, the overall correlation between differences in gene depletion FCs between cell lines with and without a specified molecular feature was equal to 0.763, and 99.2% of associations had the same sign of differential dependency across the two studies (Fig. 3a). This indicates that the studies agree not only on the existence of specific biomarkers but also on their robustness.

Gene dependency associations identified with both data sets included expected as well as potentially novel hits. Examples of expected associations included increased dependency on *ERBB2* in *ERBB2*-amplified cell lines, increased dependency on beta-catenin in APC mutant cell lines and increased dependency on *MYCN* in peripheral nervous system cell lines. A potentially novel association between *FAM72B* promoter hypermethylation and beta-catenin was also consistently identified across data sets (Fig. 3c).

We also considered gene expression to mine for possible biomarkers of gene dependency using RNA-seq data sets maintained at Broad and Sanger institutes. To this aim, we considered as potential biomarkers 1,987 genes from intersecting the top 2,000 most variable gene expression levels measured by either institute. Clustering the RNA-seq profiles revealed that each cell line' transcriptome matched closest to its counterpart from the other institute (Supplementary Fig. 4a).

We correlated the gene expression level for the most variably expressed genes to the gene dependency profiles of the SSD genes. Systematic tests of each correlation identified significant associations between gene expression and dependency. Further, as with the genomic biomarkers, we found significant overlap between gene expression biomarker associations identified in each data set

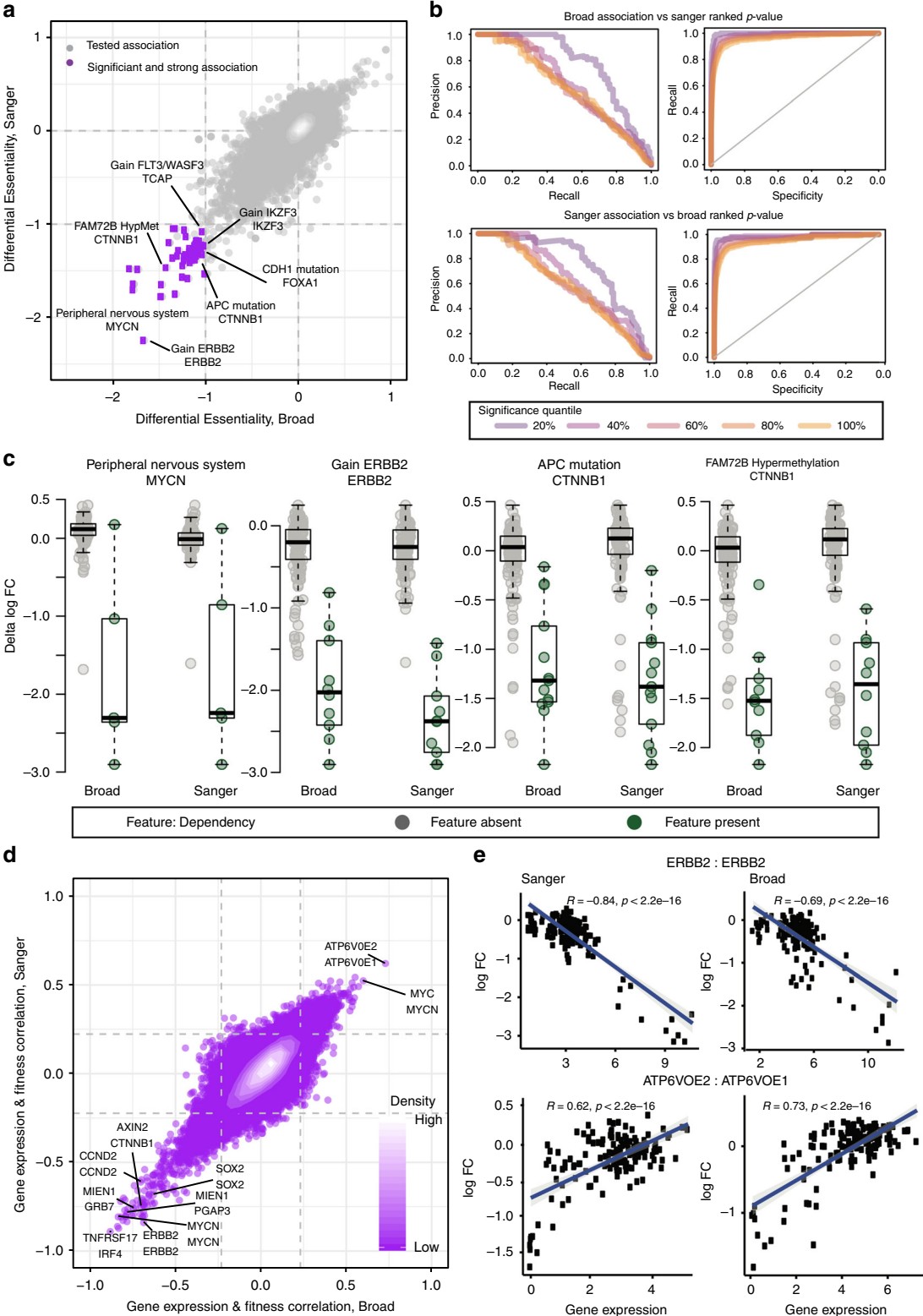

with 4,459 (52% of Broad and 66% of Sanger gene expression biomarkers) found significant for both studies, out of 97,363 tested (Fisher's exact test $p$-value below machine precision), and strong overall agreement of correlation scores between gene expression markers and SSD genes dependency across data sets

(Pearson's correlation 0.804, Fig. 3d). We observed both positive and negative correlations consistently across data sets; for example, *ERBB2* gene score was positively correlated with its expression, while *ATP6V0E1* showed significant dependency when its paralog *ATP6V0E2* had a low expression (Fig. 3e).

**Fig. 3 Reproducibility of biomarkers. a** Results from a systematic association test between molecular features and differential gene dependencies (of the SSD genes) across the two studies. Each point represents a test for differential dependency on a given gene (on the second line of the point label) based on the status of a molecular feature (on the first line). **b** Precision/Recall and Recall/Specificity curves obtained when considering as positives controls the top significant molecular-feature/gene-dependency associations found in one of the studies and ranking all the tested molecular-feature/gene-dependency associations based on their p-values in the other study. To define top-significant associations different significance thresholds matching the quantile threshold specified in the legend are considered, where 100% includes all associations with FDR less than 5%. **c** Examples of significant statistical associations between genomic features and differential gene dependencies across the two studies. The box covers the interquartile range with the median line drawn within it. The whiskers of the boxplot extend to a maximum of 1.5 times the size of the interquartile range. **d** Comparison of results of a systematic correlation test between gene expression and dependency of SSD genes across the two studies. The gray dashed lines indicate the thresholds of significant correlations at a 5% false discovery rate identified for each study. Labeled points show the gene expression marker on the first line and gene dependency on the second line. Each tested association between gene expression and SSD dependency is represented by a single purple point. Regions with higher density of points are shown in white. **e** Examples of significant correlations between gene expression and dependencies consistently identified in both studies.

**Elucidating sources of disagreement between the two data sets.** Despite the concordance observed between the Broad and Sanger data sets, we found batch effects in the unprocessed data both in individual genes and across cell lines. Although the bulk of these effects are mitigated by applying an established correction procedure[27], their cause is an important experimental question. We conducted gene set enrichment analysis of genes sorted according to the loadings of the first two principal components of the combined unprocessed gene scores using a comprehensive collection of 186 KEGG pathway gene sets from Molecular Signature Database (MsigDB)[28]. We found significant enrichment for genes involved in spliceosome and ribosome in the first principal component, indicating that screen quality likely explains some variability in the data (Supplementary Fig. 5a, b). We then enumerated the experimental differences between data sets (Fig. 1a) to identify likely causes of batch effects. The choice of sgRNA can significantly influence the observed phenotype in CRISPR-Cas9 experiments, implicating the differing sgRNA libraries as a likely source of batch effect[29]. Additionally, previous studies have shown that some gene inactivations results in cellular fitness reduction only in lengthy experiments[11]. Accordingly, we selected the sgRNA library and the time point of viability readout for primary investigation as causes of major batch effects across the two compared studies.

To elucidate the role of the sgRNA library, we examined the data at the level of individual sgRNA scores. The correlation between fold change patterns of reagents targeting the same gene (co-targeting) across studies was related to the selectivity of that gene's dependency (as quantified by the NormLRT score[21], Fig. 4a): a reminder that most co-targeting reagents show low correlation because they target genes exerting little phenotypic variation. However, even among SSDs there was a clear relationship between sgRNA correlations within and between data sets (beta test $p = 4.9 \times 10^{-10}$, $N = 49$; Fig. 4b). In particular, we note that the five SSDs (*ABHD2*, *CDC62*, *HIF1A*, *HSPA5*, *C17orf64*) identified earlier as having poor agreement between data sets have poor sgRNA correlation within data sets, thus indicating that this metric can be used to assess the reliability of a selective dependency.

One possible explanation of gene score disagreement is that sgRNAs in one of the two data sets had poor on-target efficacy. To identify such cases, we need an independent assessment of sgRNA efficacy. We estimated the efficacy of each sgRNA in both libraries using Azimuth 2.0 (ref. [29]), which uses only information about the genome in the region targeted by the sgRNA. We found that among genes identified as common dependencies in either data set, mean sgRNA depletion indeed had a strong relationship to the sgRNA's Azimuth estimated efficacy (Fig. 4c). Thus, for genes where Azimuth estimates are quite different between data sets, observed phenotype differences are probably due to differences in sgRNA efficacy. For each gene in each library, we

calculated the median estimated sgRNA efficacy (MESE) and found cases where differing MESE values appear to explain gene score differences. Some examples of this effect are *EIF3F* (common essential in Sanger screens with MESE 0.613, non-scoring in Broad screens with MESE 0.398) and *MDM2* (strongly selective in Broad screens with MESE 0.585, correlated but not strongly selective in Sanger screens with MESE 0.402) (Fig. 4d).

We next investigated the role of different experimental time points on the screens' agreement. Given that the Broad used a longer assay length (21 days versus 14 days) we expected differences to be observed between late dependencies across the data sets. Therefore, we compared the distribution of gene scores for genes known to exert a loss of viability effect upon inactivation at an early- or late-time (early or late dependencies)[11]. While early dependencies have similar score distributions in both data sets (median average score −0.781 at the Sanger and −0.830 at the Broad), late dependencies are more depleted at the Broad with median average score −0.402 compared to −0.269 for the Sanger screens (Fig. 5a). The probability of observing a difference at least this extreme for a random set of genes of the same size is $2.57 \times 10^{-78}$.

Many other experimental differences may also contribute to differences in reported response. For example, Lagziel et al. showed that many metabolic gene dependency profiles in Achilles are related to screening media, with e.g. asparagine synthetase (*ASNS*) notably more dependent in media lacking asparagine[30]. The Broad Institute used provider-recommended media for all Achilles screens, while the Sanger Institute adapted cells to either RPMI or a fifty-percent mix of DMEM and F12. While DMEM lacks asparagine, both RPMI and F12 contain it; thus, *ASNS* is expected to be a strong dependency only in Broad screens, and only in DMEM or other asparagine-deficient media. We confirmed this result (Fig. 5b). The difference between *ASNS* dependency in DMEM and either RPMI or DMEM:F12 in Broad screens is significant (Student's t-test $p = 1.52 \times 10^{-10}$, $N = 100$ and $p = 0.0173$, $N = 80$). In contrast, the difference between the RPMI and DMEM:F12 media conditions is not significant in either the Broad ($p = 0.961$, $N = 34$) or the Sanger ($p = 0.964$, $N = 147$). Although *ASNS* is the strongest example, it is likely that some of the differences in other metabolic genes between institutes are explained by media.

Unlike differences in sgRNA efficacy, both time point and media effects are expected to relate to the biological role of late dependencies. As the Broad Institute uses longer screens and includes a greater variety of media, Broad-exclusive dependencies are likely to contain enrichment for gene functional sets. We confirmed this by functionally characterizing, using gene ontology (GO), genes that were exclusively detected as depleted in individual cell lines (at 5% FDR), in one of the two studies, excluding genes with significantly different sgRNA efficacies between libraries. Results showed 29 GO categories significantly

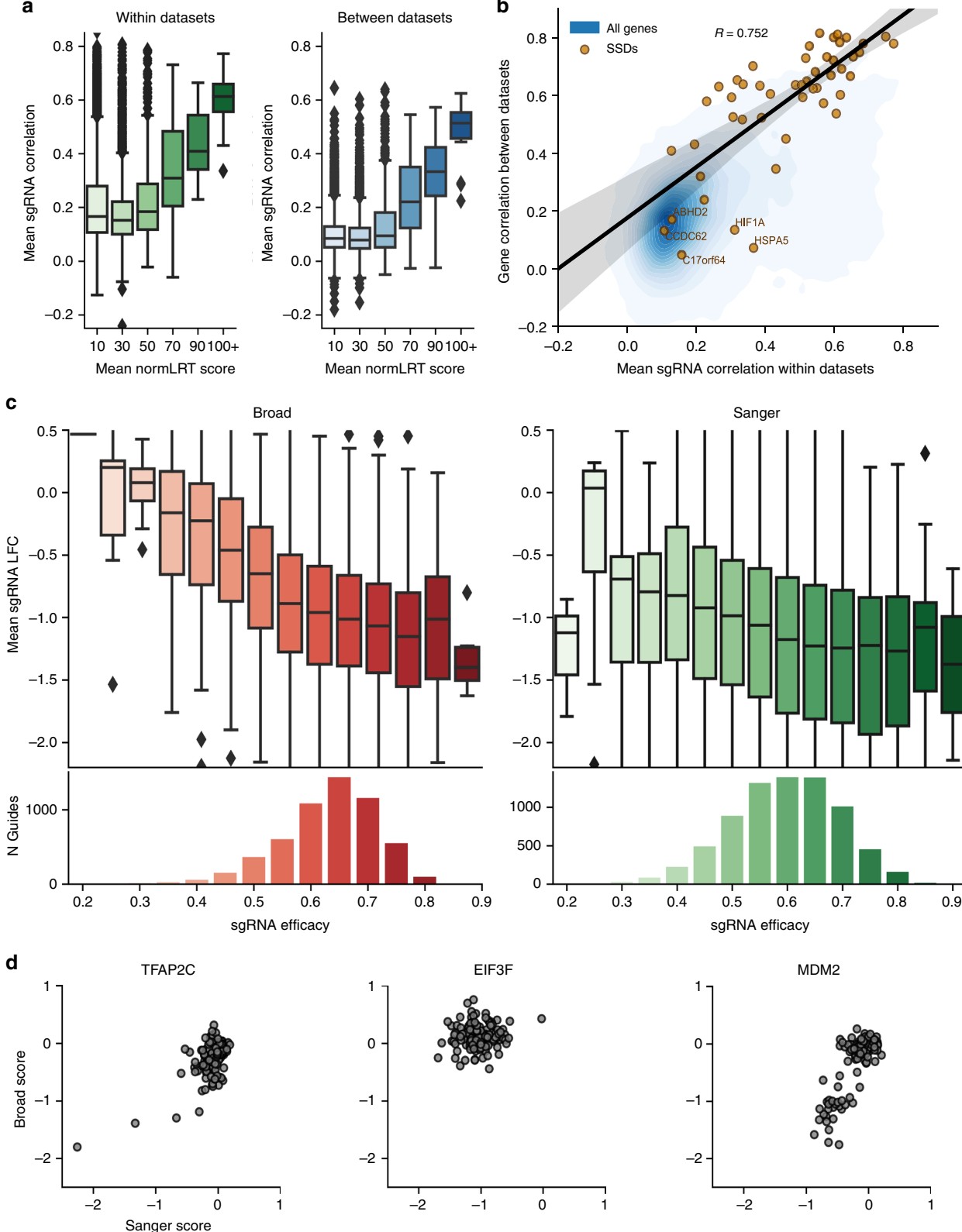

enriched in the Broad-exclusive dependency genes (Broad-exclusive GO terms) for more than 50% of cell lines (Fig. 5c and Supplementary Data 8). The Broad-exclusive enriched GO terms included classes related to mitochondrial and RNA processing gene categories and other gene categories previously characterized as late dependencies[11]. In contrast, no GO terms

were significantly enriched in the Sanger-exclusive common dependencies in more than 30% of cell lines.

**Batch effect sources: experimental verification**. To verify that batch effects between the data sets can be removed by changing

**Fig. 4 Influence of reagent library on gene score. a** Distributions of sgRNA depletion score correlations for sgRNAs targeting genes with varying NormLRT scores within each data set (left) and between them (right). Each gene is binned according to the mean of its NormLRT score across the two data sets. The *x*-axis defines the color gradient. The *y*-axis reports the average of all correlations between pairs of sgRNAs that belong to the same data set and target that gene. Boxes cover the interquartile range with the median indicated by a horizontal line. Whiskers extend up to 1.5 time the interquartile range with outliers shown as fliers. **b** Relationship between sgRNA correlation within data sets and gene correlation between data sets. The linear trend is shown for SSD genes. **c** The mean depletion of guides targeting common dependencies across all replicates vs Azimuth estimates of guide efficacy. The *x*-axis defines the color gradient. **d** Comparison of Broad and Sanger unprocessed gene scores for genes matching SSD with highest minimum median estimated sgRNA efficacy (MESE) across both libraries (left, TFA2C), common dependency in either data set and greatest difference between KY and Avana MESE (center, EIF3F), and the SSD with worst KY MESE (right, MDM2).

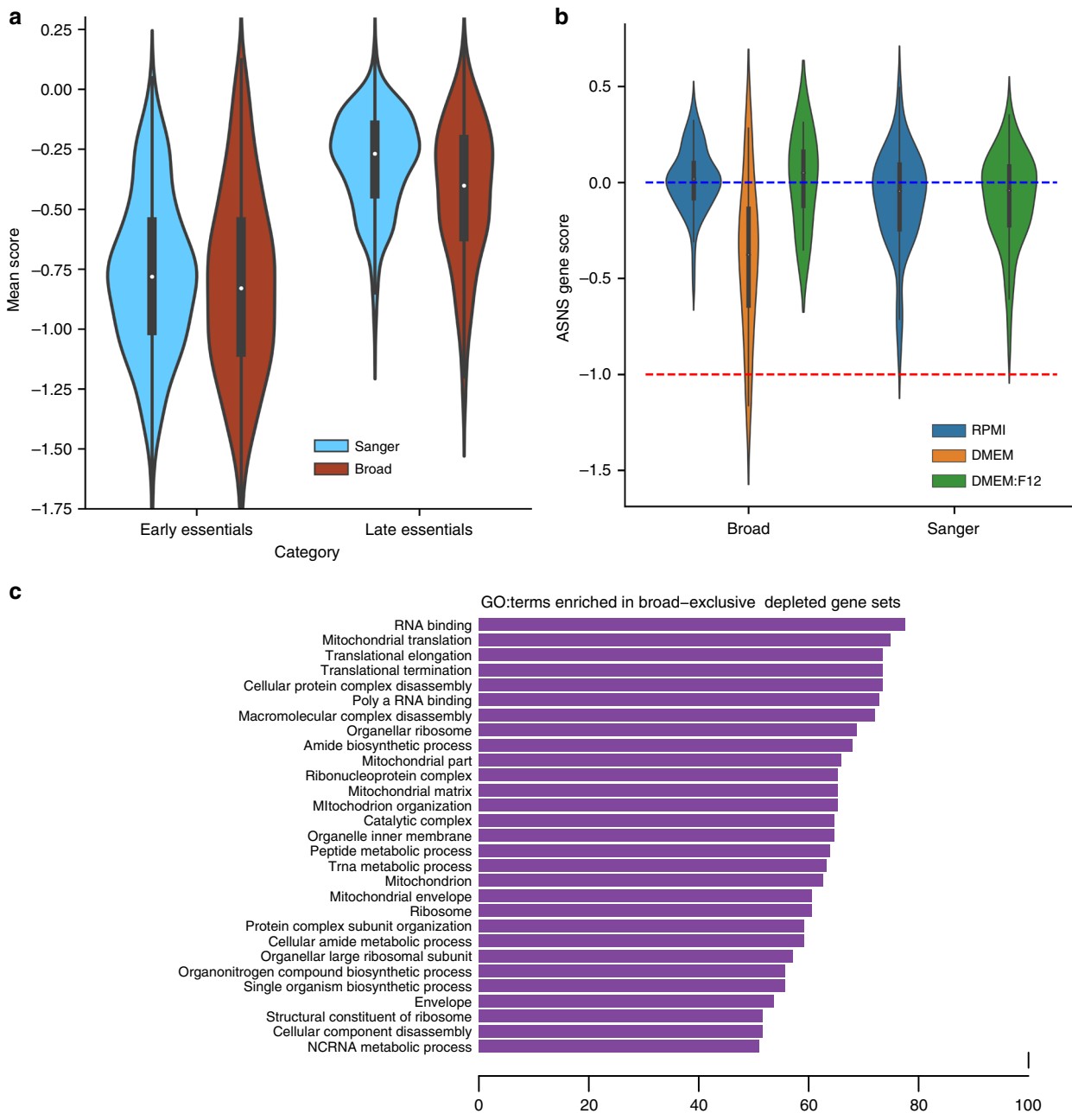

**Fig. 5 Influence of time point. a** Distribution of early and late common dependency gene scores in the Broad and Sanger data sets averaged across cell lines. Boxes cover the interquartile range with the median indicated by a horizontal line. Whiskers extend up to 1.5 time the interquartile range with outliers shown as fliers. **b** Distribution of corrected gene scores for asparagine synthetase (ASNS) by media and institute. Blue and orange lines indicate the median of nonessential and essential gene scores, respectively. **c** GO terms significantly enriched in Broad-exclusive dependencies. For each GO term the bar length indicates the ratio of cell lines showing Broad-exclusive dependencies with a statistically significant enrichment of that GO term.

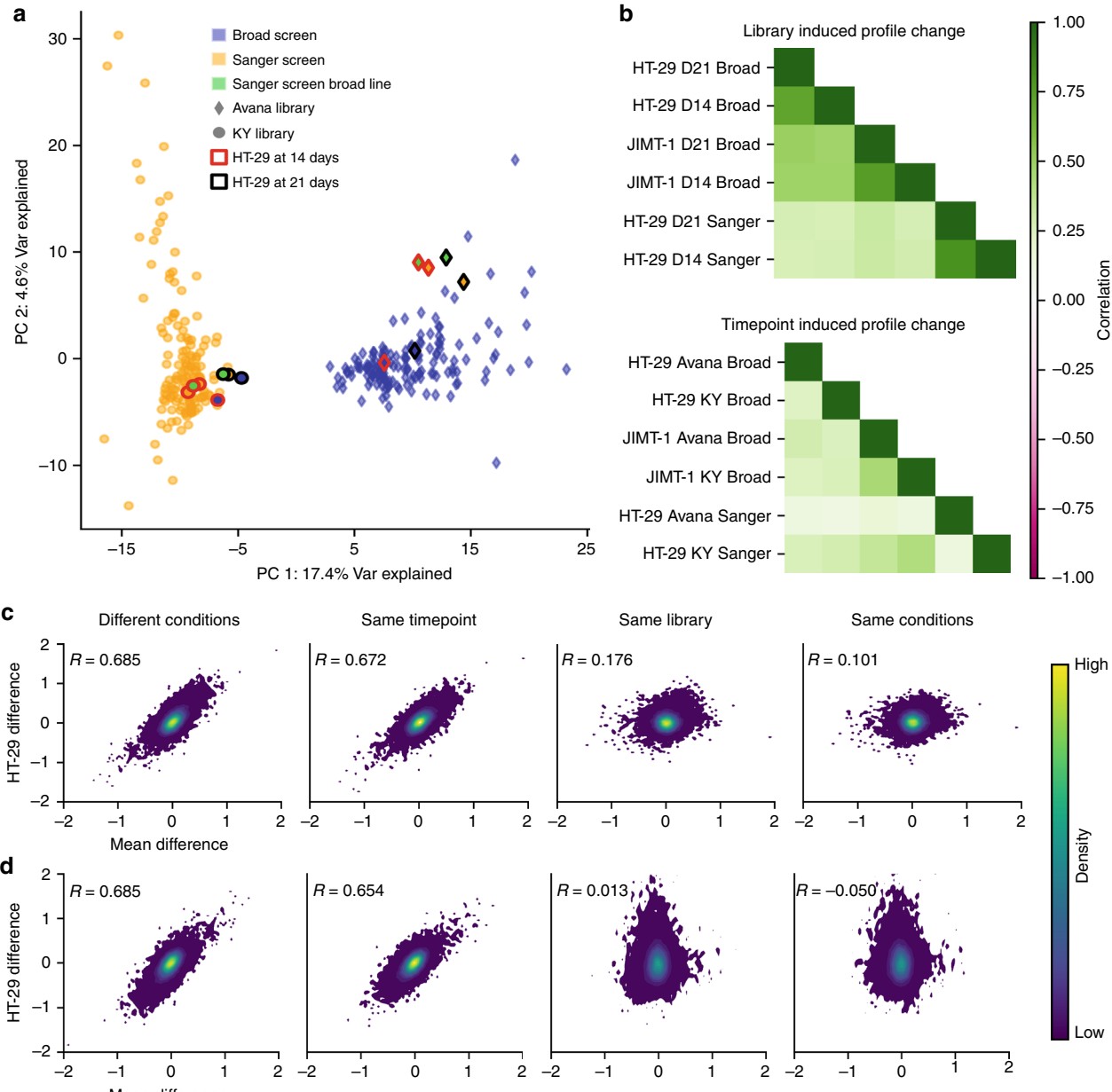

**Fig. 6 Results of replication experiments. a** Original and replication screens from each institute plotted by their first two principal components. HT-29 screens are highlighted. Axes are scaled to the variance explained by each component. **b** Correlations of the changes in gene score caused when changing a single experimental condition. **c** The difference in unprocessed gene scores between Broad screens of HT-29 and the original Sanger screen (Sanger minus Broad), beginning with the Broad's original screen and ending with the Broad's screen using the KY library at the 14-day time point. Each point is a gene. The horizontal axis is the mean difference of the gene's score between the Sanger and Broad original unprocessed data sets. **d** A similar plot taking the Broad's original screen as the fixed reference and varying the Sanger experimental conditions (Broad minus Sanger).

the library and the readout time point, we undertook replication experiments independently at Broad and Sanger institutes, where these factors were systematically permuted. The Broad sequenced cells collected from its original HT-29 and JIMT-1 screens at the 14-day time point and conducted an additional screen of these cell lines using the KY1.1 library with readouts at days 14 and 21. The Sanger used both the Broad's and the Sanger's clones of HT-29 to conduct a new KY screen and an Avana screen with readouts at days 14 and 21. Principal component analysis (PCA) of the concatenated unprocessed gene scores, including replication screens, showed a clear institute batch effect dominating the first principal component. By highlighting replication screens, we found that this effect is chiefly due to library choice, with time

point playing a smaller role (Fig. 6a, Supplementary Fig. 6a). Changing from Sanger to Broad clones of HT-29 had minimal impact. We examined the change in gene score profile for each screen caused by changing either the library or time point while keeping other conditions constant. Gene score changes induced by either library or time point alterations were consistent across multiple conditions (Fig. 6b). Sanger-exclusive common dependencies were strongly enriched for genes that became more depleted with the KY library, and Broad-exclusive common dependencies were enriched among genes more depleted with the Avana library (Supplementary Fig. 6b). Late dependencies were strongly enriched among genes that became more depleted in the later time points, while early dependencies were not

(Supplementary Fig. 6c). We compared the deviations in gene score between Broad and Sanger screens under different conditions, first comparing Broad original and replication screens of HT-29 (Fig. 6c) and JIMT-1 (Supplementary Fig. 6d) to the original Sanger screens of the same cell line. Matching library and time point removed most of the average gene score change (batch effect) between institutes, as indicated by the low correlation of the remaining gene score differences in the replication screens with the average gene score change. Specifically, matching Sanger's library and time point reduces the variance of gene scores in HT-29 from 0.0486 to 0.0252 and in JIMT-1 from 0.0556 to 0.0260. We next compared Sanger original and replication screens of HT-29 to the Broad original HT-29 screen. Matching library and time point successfully detrended the data in this case as well; however, the Sanger Avana screens of HT-29 contained considerable excess noise, causing these screens to have a higher overall variance from the Broad than the original screens (0.0486 vs 0.115). Nonetheless, the replication experiments confirm that the majority of batch effects between data sets are driven by the library and time point.

## Discussion

Providing sufficient experimental data to adequately sample the diversity of human cancers requires high-throughput screens. However, the benefits of large data sets can only be exploited if the underlying experiments are reliable and robustly reproducible. In this work, we survey the agreement between two large, independent CRISPR-Cas9 knock-out data sets, generated at the Broad and Sanger institutes.

Our findings illustrate a high degree of consistency in estimating gene dependencies between studies at multiple levels of data processing, albeit with the longer duration of the Broad screens leading to stronger dependencies for a number of genes. The data sets are concordant in identifying common dependencies and identifying mean dependency signals. Their agreement is also striking in the more challenging task of identifying which cell lines are dependent on selective dependencies. Indeed, when we compared the two data sets at the level of gene dependency markers we found consistent results at the level of common informative molecular features, as well as with respect to their quantitative strength.

We observed that a source of disagreement across the compared data set is due to diffuse batch effects visible when the whole profiles of individual cell lines are compared. Such effects can be readily corrected with standard methods without compromising data quality, thus making possible integration and future joint analyses of the two compared data sets. Furthermore, much of this batch effect can be decomposed into a combination of two experimental choices: the sgRNA library and the duration of the screen. The effect of each choice on the mean depletion of genes is readily explicable and reproducible, as shown by screens of two lines performed at the Broad using the Sanger's library and screen duration and a reciprocal screen performed at the Sanger with the Broad library and duration. Consequently, identifying high-efficacy reagents and choosing the appropriate screen duration should be given high priority when designing CRISPR-Cas9 knock-out experiments.

## Methods

**Unprocessed gene scores**. Read counts for the Broad were taken from avana_-public_19Q1 (ref. [31]) and filtered so that they contained only replicates corresponding to overlapping cell lines and only sgRNAs with one exact match to a gene. Read counts for Sanger were taken from Behan et al.[13] and similarly filtered, then both read counts were filtered to contain only sgRNAs matching genes common to all versions of the data. In both cases, reads per million (RPM) were calculated and an additional pseudo-count of 1 added to the RPM. Log fold change was calculated from the reference pDNA. In the case of the Broad, both pDNA and

screen results fall into distinct batches, corresponding to evolving PCR strategies. Cell lines sequenced with a given batch were matched to pDNA profiles belonging to the same batch. Multiple pDNA RPM profiles in each batch were median-collapsed to form a single profile of pDNA reads for each batch. Initial gene scores for each replicate were calculated from the median of the sgRNAs targeting that replicate. Each replicate's initial gene scores for both Broad and Sanger were then shifted and scaled so the median of nonessential genes in each replicate was 0 and the median of essential genes in each replicate was negative one[12]. Replicates were then median-collapsed to produce gene- by cell-line matrices.

**Processed gene scores**. Broad gene scores were taken from avana_public_19Q1 gene_effect[31] and reflect CERES[15] processing. The scores were filtered for genes and cell lines shared between institutes and with the unprocessed data, then shifted and scaled so the median of nonessential genes in each cell line was 0 and the median of essential genes in each cell line was −1 (ref. [12]). Sanger gene scores were taken from the quantile-normalized averaged log fold-change scores, post-correction with CRISPRcleanR[32], and globally rescaled by a single factor so that the median of essential genes across all cell lines was −1 (ref. [12]).

**Batch-corrected gene scores**. The unprocessed sgRNA log FCs were mean collapsed by gene and replicates. Data were quantile normalized for each institute separately before processing with ComBat using the R package sva. One batch factor was used in ComBat defined by the institute of origin. The ComBat corrected data were then quantile normalized to give the final batch-corrected data set.

**Alternate conditions**. Screens with alternate libraries, cell lines, and time points were processed similarly to the Unprocessed data above.

**Gene expression data**. Gene expression log2(Transcript per million +1) data were downloaded for the Broad from the Figshare repository for the Broad data set. For the Sanger data set, we used fragments per kilobase million (FPKM) expression data from Cell Model Passports[33]. We added a pseudo-count of 1 to the FPKM values and transformed to log2. Gene expression values are quantile normalized for each institute separately. For the Sanger data, Ensembl gene ids were converted to Hugo gene symbols using BiomaRt package in R.

**Guide efficacy estimates**. On-target guide efficacies for the single-target sgRNAs in each library were estimated using Azimuth 2.0 (ref. [29]) against GRCh38.

**Comparison of all gene scores**. Gene scores from the chosen processing method for both Broad and Sanger were raveled and Pearson correlations calculated between the two data sets. 100,000 gene-cell line pairs were chosen at random and density-plotted against each other using a Gaussian kernel with the width determined by Scott's rule[34]. All gene scores for essential genes were similarly plotted in Fig. 1b.

**Comparison of gene means**. Cell line scores for each gene in both Broad and Sanger data sets with the chosen processing method were collapsed to the mean score, and a Pearson correlation calculated.

**Gene ranking, common essential identification**. For each gene in the chosen data set, its score rank among all gene scores in its 90th percentile least depleted cell line was calculated. We call this the gene's 90th percentile ranking. The density of 90th percentile rankings was then estimated using a Gaussian kernel with width 0.1 and the central point of minimum density identified. Genes whose 90th percentile rankings fell below the point of minimum density were classified as common essential.

**Identification of selective gene sets**. Selective dependency distributions across cell lines are identified using a Likelihood Ratio Test as described in McDonald et al.[21]. For each gene, the log-likelihood of the fit to a normal distribution and a skew-t distribution is computed using R packages MASS[35] and sn[36], respectively. In the event that the default fit to the skew-t distribution fails, a two-step fitting process is invoked. This involves keeping the degrees of freedom parameter (ν) fixed during an initial fit and then using the parameter estimates as starting values for a second fit without any fixed values. This process repeats up to 9 times using ν values in the list (2, 5, 10, 25, 50, 100, 250, 500, 1000) sequentially until a solution is reached. The reported LRT score is calculated as follows:

$$LRT = 2 * [\ln(\text{likelihood for Skewed} - t) - \ln(\text{likelihood for Gaussian})] \quad (1)$$

The numerical optimization methods used for the estimates do not guarantee the maximum of the objective function is reached. In a small number of cases, we failed to find a solution even with multiple attempts. NormLRT scores have been left blank for these genes. Genes with NormLRT scores greater than 100 and mean gene score greater than −0.5 in at least one institute's unprocessed data set were classified as SSDs.

**Binarized agreement of SSDs**. For each processing method, Broad and Sanger gene scores were concatenated. Scores for nonessential genes across all cell lines and both institutes were taken as the null distribution, and a left-tailed $p$-value calculated for each score. The resulting $p$-values for each processing method were converted to FDR using the Benjamini–Hochberg algorithm as implemented in the python package statsmodels. The gene score threshold corresponding to a FDR of 0.05 or lower was used to binarize gene scores. These thresholds were $-1.02$ for unprocessed gene scores, $-0.633$ for processed gene scores, and $-0.765$ for corrected gene scores. Cohen's kappa was calculated for each gene individually. Fisher's exact test, precision, recall, and AUROC scores were calculated globally for all SSD sensitivities in the three data versions.

**Cell line agreement analysis**. To obtain the two dimensional visualisations of the combined data set before and after batch correction and considering different gene sets, we computed the sample-wise correlation distance matrix and used this as input into the t-statistic Stochastic Neighbor Embedding (tSNE) procedure[25], using the *tsne* function of the tsne R package, with 1000 iterations, a perplexity of 100 and other parameters set to their default value.

To evaluate genome-wide cell line agreement we considered a simple nearest-neighbor classifier that, for each dependency profile of a given cell line in one of the two studies, predicted its matching counterpart in the other study. This prediction was based on the correlation distance between one profile and all the other profiles. To estimate the performance of this classifier, we computed a Recall curve for each of the 294 dependency profiles in the tested data set. Each of these curves was assembled by concatenating the number of observed true-positives amongst the first $k$ neighbors of the corresponding dependency profile (for $k = 1$–293). We then averaged the 294 resulting Recall curves into a single curve and converted it to percentages by multiplying by 100/294. Finally, we computed the area under the resulting curve and normalized it by dividing by 293. We considered the area under this curve (nAUC) as a performance indicator of the k-nearest neighbor.

**Cell line profiles agreement in relation to data quality**. First, to estimate the initial data quality we calculated true positive rates (TPRs, or Recalls) for the sets of significant dependency genes detected across cell lines, within the two studies. To this aim, we used as positive control a reference set of a priori known essential genes[12]. We assessed the resulting TPRs for variation before/after batch correction, and for correlations with the inter-study agreement.

**Biomarker analysis**. We used cell lines' binary event matrices based on mutation data, copy number alterations, the tissue of origin and MSI status. The resulting set of 587 features were present in at least 3 different cell lines and fewer than 144. We performed a systematic two-sample unpaired Student's $t$-test (with the assumption of equal variance between compared populations) to assess the differential essentiality of each of the SSD genes across a dichotomy of cell lines defined by the status (present/absent) of each CFE in turn. SSD genes were those with NormLRT values greater than 100 in either institute. From these tests, we obtained $p$-values against the null hypothesis that the two compared populations had an equal mean, with the alternative hypothesis indicating an association between the tested CFE/gene-dependency pair. $P$-values were corrected for multiple hypothesis testing using Benjamini–Hochberg. We also estimated the effect size of each tested association by means of Cohen's Delta ($\Delta$FC), i.e. difference in population means divided by their pooled standard deviations. For gene expression analysis we calculated the Pearson correlation across the cell lines between the SSD gene dependency profiles and the gene expression profiles from each institute. The significance of the correlation was assessed using the t-distribution ($n - 2$ degrees of freedom) and $p$-values were corrected for multiple hypothesis testing using the q-value method.

For the agreement assessment via ROC indicators (Recall, Precision and Specificity), for each of the two studies in turn we picked the most significant 20, 40, 60, 80, and 100% associations as true controls and evaluated the performance of a rank classifier based on the corresponding significance p-values obtained in the other study.

For the analysis involving transcriptional data, we used the RNA-seq data from each institute for overlapping cell lines, which includes some sequencing files that have been used by both institutes and processed separately.

**Rank-based dependency significance and agreement**. To identify significantly depleted genes for a given cell line, we ranked all the genes in the corresponding essentiality profiles based on their depletion logFCs (averaged across targeting guides), in increasing order. We used this ranked list to classify genes from two sets of prior known essential ($E$) and non-essential ($N$) genes, respectively[12].

For each rank position $k$, we determined a set of predicted genes $P(k) = \{s \in E \cup N : \varrho(s) \leq k\}$, with $\varrho(s)$ indicating the rank position of $s$, and the corresponding precision $PPV(k)$ as:

$$PPV(k) = |P(k) \cap E| / |P(k)|$$

Subsequently, we determined the largest rank position $k^*$ with $P(k^*) \geq 0.95$ (equivalent to a FDR $\leq 0.05$). Finally, a 5% FDR logFCs threshold $F^*$ was determined as the logFCs of the gene s such that $\varrho(s) = k^*$, and we considered all the genes with a logFC $< F^*$ as significantly depleted at 5% FDR level. For each cell

line, we determined two sets of significantly depleted genes (at 5% FDR): $B$ and $S$, for Broad and Sanger data sets, respectively. We then quantified their agreement using the Jaccard index[37] $J(B, S) = |B \cap S| / |B \cup S|$, and defined their disagreement as $1 - (B, S)$. Summary agreement/disagreement scores were derived by averaging the agreement/disagreement across all cell lines.

**sgRNA correlations**. Broad and Sanger log fold-changes for their original screens were median-collapsed to guide by cell line matrices. For each gene present in the unprocessed gene scores, a correlation matrix between all the sgRNAs targeting that gene in each guide by cell line matrix was computed. The mean of the values in this matrix for each institute, excluding the correlations of sgRNAs with themselves, was retained. The mean sgRNA correlation within institutes was then calculated from the mean of the Broad and Sanger sgRNA correlation matrix means. The mean sgRNA correlation between institutes for each gene was calculated from the mean of all possible pairs of sgRNAs targeting that gene with one sgRNA chosen from Sanger and one from Broad.

**Relating sgRNA depletion and efficacy**. We chose the set of genes found to be essential in at least one unprocessed data set. The log fold-change of guides targeting those genes in each data set was calculated and compared to the guide's estimated on-target efficacy.

**Difference in late essential gene scores between data sets**. We randomly selected $n$ genes, where $n$ is the number of late essential genes, and calculated the difference in median gene score for those genes between the Broad and Sanger institutes. We repeated this 10,000 times to generate the null distribution for median difference. No instances of the null were as extreme as the observed difference between median late essential scores. However, the null was well-approximated by a Gaussian distribution, which allowed us to extrapolate a $p$-value for the observed difference in medians.

**Time point gene ontology analysis**. We tested for enrichment of GO terms associated with genes showing a significant depletion in only one institute. To rule out the differences due to the library, genes with significantly different guide efficacies were filtered from the analysis. Using the Azimuth scores average (mean) efficacy scores for each gene at each institute were calculated. A null distribution of differences in gene efficacy was estimated using genes not present in either institute specific sets (which were defined as depleted in at least 25% of cell lines). Institute specific genes greater than 2 standard deviations from the mean of the null distribution were removed.

For the filtered gene set prior known essential and non-essential gene sets from[32] were used to find significant depletions for each cell line and institute at 5% FDR. For each cell line, the genes identified as significantly depleted in only Broad or only Sanger were functionally characterized using GO enrichment analysis[38]. To this aim, we downloaded a collection of gene sets (one for each GO category) from the Molecular Signature Database (MsigDB)[28], and performed a systematic hypergeometric test to quantify the over-representation of each GO category for each set of study-exclusive dependency genes, per cell line. We corrected the resulting $p$-values for all the tests performed within each study using the Benjamini–Hochberg procedure[39], and considered a GO category enriched in a cell line if the corrected $p$-value resulting from the corresponding test was $< 0.05$.

**Principal component analysis of the batch effect**. The Broad and Sanger unprocessed gene scores and the gene scores for the alternate conditions tested by both institutes were concatenated into a single matrix with a column for each screen. Principal components were found for the transpose of this matrix, where each row is a screen and each column a pseudogene. Components 1 and 2 were plotted for all original screens and the alternate screens for either HT-29 (Fig. 6a) or JIMT-1 (Supplementary Fig. 6a). The aspect ratio for the plot was set to match the relative variance explained by the first two principal components.

**Consistency of time point and library effects on gene scores**. To evaluate library differences, we took all screens that had been duplicated in each library with all other conditions (time point, clone, and screen location) kept constant. For each of these screens, we subtracted the gene scores of the version performed with the KY library from the version performed with the Avana library to create library difference profiles. For the case of Sanger's day-14 KY screen of the Sanger HT-29 clone, two versions exist, the original and an alternative that was eventually grown out to 21 days. We used the alternate version of this screen to be consistent with the day 21 results. A correlation matrix of library difference profiles was then calculated and is plotted to the left of Fig. 6b. The procedure was repeated for time point differences, creating time point difference profiles by subtracting day 14 results from day 21 results for pairs of screen readouts that differed in time point but not library, clone, or screen location.

**Matching experimental conditions**. For the cell line HT-29, we took Sanger's original screen as a baseline. We then subtracted from this baseline from four Broad HT-29 screens: the original (Avana library at day 21), then with the Avana

library at day 14, the KY library at day 21, and the KY library at day 14, generating four arrays indexed by gene which form the y-axes in the succession of plots in Fig. 6c. We also computed the mean score of each gene across all original Broad screens and subtracted it from the mean score of each gene across all the original Sanger screens to form the x-axis of all four plots. For each condition, the standard deviation of the HT-29 screen differences (y-axes) was computed along with the correlation of the HT-29 screen differences with the mean differences (x-axis). The plots themselves are Gaussian kernel density estimates. We repeated this process for JIMT-1 (Supplementary Fig. 6d) and then for HT-29 while swapping the roles of Broad and Sanger (Fig. 6d). For the Sanger alternate condition screens we used the Sanger clone of HT-29, and for its day 14 KY screen we used the Sanger's original HT-29 screen.

**Replication experiments.** The replication screens at Broad and Sanger were performed using the normal current protocol of the respective institution[13,15] except with respect to the specifically noted changes to the library (and the associated primer sequences required for post-screen amplification of the sgRNA barcodes) and the time point. See Supplementary Methods for details.

**Reporting summary.** Further information on research design is available in the Nature Research Reporting Summary linked to this article.

## Data availability

The data used for this paper have been posted to Figshare (https://doi.org/10.6084/m9.figshare.7970993.v1).

## Code availability

Scripts to perform all analyses and generate figures are available at https://github.com/DepMap-Analytics/Comparative-Analysis.

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

## Acknowledgements

This work was funded by Open Targets (OTAR2-055) to F.I and (OTAR015) to M.J.G. and K.Y., by the Wellcome Trust grant no. 206194 to M.J.G., by Wellcome and the Estonian Research Council (IUT 34-4) to L.P., by grants U01 CA176058 and U01 CA199253 to W.C.H and by the HL Snyder Foundation (W.C.H.).

## Author contributions

J.M.D., F.I. and A.T. conceived and designed the study. J.M.D. and C.P. conducted the analyses described under Results. J.M.D., C.P. and F.I. wrote the paper and produced the figures. A.T. wrote the paper. H.N. produced figures and curated data. J.M.D. munged and collated gene scores. C.P. munged and collated cell characterizations. J.K.-.B. produced the script used to calculate NormLRT scores. V.Z., S.P., S.T.Y. and D.E.R. conducted the Broad's replications of Sanger screens, while F.B., R.S. and C.M.B conducted Sanger's replications and curated corresponding data. J.G.D. and K.Y. provided ideas and discussed the integration of Avana and KY libraries, and T.G. provided the Azimuth scores for both. F.A., E.G., F.V., L.P., J.S.B., T.R.G., W.C.H. and M.J.G. edited the paper and contributed ideas on some of the analyses. J.S.B., T.R.G., W.C.H. and M.J.G. acquired funds and contributed to study supervision. A.T. and F.I. acquired funds and supervised the study.

## Competing interests

C.P., F.M.B., H.N., M.J.G. and F.I. receive funding from Open Targets, a public-private initiative involving academia and industry. K.Y. and M.J.G. receive funding from AstraZeneca. M.J.G performed consultancy for Sanofi. J.G.D. and A.T. perform

consulting for Tango Therapeutics. W.C.H. performs consulting for Thermo Fisher, AdjulB, MBM Capital, and Paraxel, and is a founder and scientific advisory board member of KSQ Therapeutics. T.R.G. performs consulting for GlaxoSmithKline, Sherlock Biosciences, and Foundation Medicine. F.I. performs consultancy for the joint CRUK - AstraZeneca Functional Genomics Centre. All the other authors declare no competing interests.
