## [Peer Review File · Nature Communications]

Reviewers' comments:

Reviewer #1 (Remarks to the Author):

Dempster et. al perform an analysis to compare the consistency of two large pan-cancer CRISPR-Cas9 dependency screening datasets (one dataset generated at the Sanger Institute, one at the Broad, 147 overlapping cell lines).

The major conclusion of the paper is that the data generated at these two centers are reasonably consistent.

This is the first analysis of its kind and it is an important analysis. Its importance stems from the previous controversy around the consistency of a drug screening data generated on two similar large panels of cell lines generated at these two institutes. This previous debacle was very time consuming for many people and hurt the progress of cancer research, so I believe it is wise of these authors to perform such a study at this point in time.

The major problem with the assessments of consistency in drug screening data was (as the authors point out again in this manuscript) that a lack of covariance between repeated measurements can result from a lack of agreement, but also a lack of variance. In this manuscript the authors have wisely pointed this out, which will hopefully prevent future problems with the interpretation of these data and their consistency.

In this manuscript, the authors have clearly demonstrated that when variability is present (i.e. for genes where there is selective dependence), there is a strong correlation (i.e. covariance) between the two datasets. As expected, when there is no variance, there is little covariance.

In general, this is an important study, the statistical approaches are sound and there is little need for additional analysis to strengthen conclusions.

However, I have some very minor comments/questions:

- The choice of ComBat is interesting. This tool was designed for DNA microarrays, and works by standardizing the mean and variance of each gene between the two datasets. Is there some evidence that in this scenario, simply standardizing the mean and variance might work just as well or better? The empirical Bayes step employed by ComBat is designed for small sample sizes and is probably not doing much when you have 147 cell lines. Its also arguably more confusion for the causal reader.

- The improved Pearson's correlations reported in section 2 of results are probably just because the data have been forced onto a linear scale. I.e. I'm guessing there's little improvement in the Spearman's correlation? Which is probably worth reporting?

- The Pearson's correlation of 0.9997 reported and shown in Supplementary Figure 1a couldn't possibly be biologically meaningful (unless I'm very much mistaken) and looks to be an artifact of the normalization procedure, thus this number is pretty meaningless? Is it worth reporting Spearman's correlation here again?

- Perhaps an issue of personal taste but some of the numbers written on some of the figures look a little strange. E.g. fig. 1(d). Maybe worth writing something like "n=396" instead of "396" and aligning these consistently in the different quadrants?

In conclusion, I think this paper needs to be published. The findings are important, the approaches are generally solid, and I think it will save a lot of people a lot of headaches moving forward.

Reviewer #2 (Remarks to the Author):

Dempster and colleagues have performed a comprehensive evaluation of the agreement between the two largest CRISPR-Cas9 gene loss-of-function screenings in cancer cell lines. The findings derived from this evaluation represent an important reference for the scientific community. Many research groups plan to use this valuable data from the Cancer Dependency Map project. Thus, understanding how reproducible are read-out from this platform is fundamental for the scientific community.

The current status of the manuscript is very good and of high quality. The article is well written and well organized. The statistical tests performed to address the different comparisons are adequate. The different levels of comparisons between the two data sets are comprehensive.

Major point:

The whole study is focused in showing the level of agreement between the two datasets. In contrast the source of disagreement is attributed mainly to technical variation (batch effect): reagents, screening quality and time point of read-out which was corrected it using Combat. However, it would be expected a certain level of disagreement due to biological variability as well. For instance, it would be expected that not all but key essential metabolic genes could disagree between the two datasets if the media culture used to grow the cells in each screening had different composition of essential metabolites. The authors showed the source of technical variation using PCA and visual inspection but a more careful inspection of the PC loadings should be performed to dissect if it also includes some biological variation. Although they performed a broad exploration of biological variation of those hits only found in a single dataset using GO terms, specific and interesting cases might be investigated. Thus, a detailed functional analysis on the PC loadings should be performed that would help to describe the type of biological variability if any.

In summary, authors should include an extended interpretation of the biological variability to complete their study that it would help researchers to understand the scope of this data for their projects.

Minor points:

- Figure 1 represents a good depiction of the problem and how they addressed it. However, the density of the scatter plot is saturated. It makes difficult to precisely visualize the agreement between the two datasets. Taking into account that the CRISPR screenings are performed in individual cell lines, one would expect to check this out in a single sample basis. Is there any cell line for which there is no agreement at all between the two datasets? I would suggest to add a supplementary table with Pearson's correlation coefficients per each cell line that was tested in both screenings. A supplementary boxplot with single-sample gene dependency correlations between the two screenings stratified by tissue might help the research community to figure out how "homogeneous" is the agreement across tissues and cell lines.
- In the "Agreement of gene dependency biomarkers" section, the authors performed a genome-wide t-test for significant differences in dependency scores between the two cell populations. It was difficult to understand how they defined the two cell populations for the statistical test. It seems that they stratified cell lines by the presence of a feature of interest (biomarker). Authors should explain better this stratification at the beginning of the section.
- Fig 6A must have resize to be squared. Otherwise patterns are exaggerated along the rotated data from one principal component (i.e. PC1 in the plot).

Reviewer #3 (Remarks to the Author):

The manuscript by Dempster et al., provides an analysis of the reproducibility of two recent large scale CRISPR-Cas9 functional genomics studies from the Sanger and Broad Institutes. These complimentary efforts represent an extraordinary resource for the community. It is therefore completely justified and important to understand the reproducibility of these efforts but also their differences. The authors conclude that in general these CRISPR-Cas9 screening efforts are highly concordant and identify two contributing factors, library effect and assay length, to observed disagreement in gene dependency scores. While this is useful knowledge for the research community who will be mining these datasets and planning functional genomics studies, library batch effect observed that arise from factors such as different numbers gRNA per gene, variable gRNA knockout efficiency, and other experimental design have been previously recognized. However, the application of a previously described method (ComBAT) to correct for batch effect is certainly novel and a relatively simple way to integrate screens from various groups. Interestingly, while the authors are focusing on identifying similarities between the two screening efforts, one could equally pay attention to their differences and factors that could explain the variations (once the screens are normalized). I suspect that this could be the focus of a follow up study by this group or others.

Major Issues

- 1 As mentioned above, the novelty of this study could be questioned. Contributions of library composition and assay length to study-to-study variability have been previously described. Aside from a verification that these previous studies are more or less in agreement, when applied to large scale efforts. Given that these two large scale efforts will be a source of important information for the community moving forward, a study rigorously comparing the two efforts is however fully justified.
2. Figure 2B does not appear to be cited in the manuscript text.
3. Throughout the paper the terms “gene score” “gene dependency score” and “dependency score” appear to be used interchangeably. Do these all refer to the same metric? If so please clarify, and if not select one description and use throughout to increase clarity.
4. While differences in sgRNA efficiency is likely a key contributor to observed differences in screen results, I do not think the data shown in Figure 4 demonstrates this very well. Fig. 4B does a pretty good job of showing potentially problematic genes based on sgRNA correlation—however, later in the figure (panel D) a completely different set of genes is shown. Why not continue with the first set of genes? Furthermore, for Fig 3D, while the text states that “reagent efficiency likely explains some differences” no data matching this description is provided as the corresponding panel simply shows differences in gene scores. This section needs improved presentation of how reagent efficiency differences affect the scoring of example genes.

Minor Issues

1. In caption for Fig. 1B it reads “The distributions scores...” should this read “The gene scores...” or “The distribution of gene scores...”
2. In Fig. 1D it is not entirely clear that the labels used in panel 1C are meant to carry over.
3. When analyzing cell line dependencies, a threshold of <-0.7 was employed. What is the rationale for this threshold? The raw number itself will have limited meaning to most readers so more information is warranted.
4. In Fig. 2D what does the “Au” in the axis labels refer to?
5. The description of the data shown in Fig. 3B is confusing and therefore it is difficult to understand what value it adds.
6. While MYCN is depicted in Fig 3C it is not described in the text while all others shown are.
7. Fig 3D is said in the text to depict “significant associations between gene expression and dependency” which is actually quite difficult to discern from the figure panel. The panel for 3D instead shows a study to study comparison of these correlation scores.
8. Caption for Fig 4D uses the acronym “MESE” but not explanation is provided.
9. Is the difference in the mean score for late essentials between studies statistically significant?

To facilitate readability and clarity, in the following point-by-point response to reviewers' comments we have reported *original reviewers' statements in Italic*, our responses in *blue Italic*, and portions of text from the revised version of our manuscript in *Blue Courier New font*.

Reviewer #1 (Remarks to the Author):

Dempster et. al perform an analysis to compare the consistency of two large pan-cancer CRISPR-Cas9 dependency screening datasets (one dataset generated at the Sanger Institute, one at the Broad, 147 overlapping cell lines).

The major conclusion of the paper is that the data generated at these two centers are reasonably consistent.

This is the first analysis of its kind and it is an important analysis. Its importance stems from the previous controversy around the consistency of a drug screening data generated on two similar large panels of cell lines generated at these two institutes. This previous debacle was very time consuming for many people and hurt the progress of cancer research, so I believe it is wise of these authors to perform such a study at this point in time.

The major problem with the assessments of consistency in drug screening data was (as the authors point out again in this manuscript) that a lack of covariance between repeated measurements can result from a lack of agreement, but also a lack of variance. In this manuscript the authors have wisely pointed this out, which will hopefully prevent future problems with the interpretation of these data and their consistency.

In this manuscript, the authors have clearly demonstrated that when variability is present (i.e. for genes where there is selective dependence), there is a strong correlation (i.e. covariance) between the two datasets. As expected, when there is no variance, there is little covariance.

In general, this is an important study, the statistical approaches are sound and there is little need for additional analysis to strengthen conclusions.

We thank the reviewer for their positive comments.

However, I have some very minor comments/questions:

- The choice of ComBat is interesting. This tool was designed for DNA microarrays, and works by standardizing the mean and variance of each gene between the two datasets. Is there some evidence that in this scenario, simply standardizing the mean and variance might work just as well or better? The empirical Bayes step employed by ComBat is designed for small sample sizes and is probably not doing much when you have 147 cell lines. Its also arguably more confusion for the causal reader.

We agree with the reviewer that there is little practical difference between ComBat and simply standardizing means and variances in this case, although ComBat does not align variances as strictly as means, as now shown in Supplementary Fig. 1b. We opted for ComBat for its wide adoption and simplicity: it is frequently the first tool chosen for correcting biological batch effects prior integration of different datasets.

- The improved Pearson's correlations reported in section 2 of results are probably just because the data have been forced onto a linear scale. I.e. I'm guessing there's little improvement in the Spearman's correlation? Which is probably worth reporting?

If we are correctly understanding the reviewer's point, we would like to highlight that although the data does appear more linearly arranged after correction in the results showed in section 2, it has not been forced to a linear scale (except in the sense that ComBat has forced the gene score means to fall on the diagonal). Thus Spearman correlation is also improved.

We do agree with the reviewer that this is worth being reported. As a consequence, we have added the following text in the "Agreement of gene scores" section:

Spearman correlations are 0.347, 0.411, and 0.551 respectively, again significant below machine precision.

- The Pearson's correlation of 0.9997 reported and shown in Supplementary Figure 1a couldn't possibly be biologically meaningful (unless I'm very much mistaken) and looks to be an artifact of the normalization procedure, thus this number is pretty meaningless? Is it worth reporting Spearman's correlation here again?

The reviewer is correct that this is determined by the batch correction. We presented it only for completeness (and to make the point that the batch correction aligns gene means). In addition to adding a supplementary panel on gene variance agreement post-correction, we have rearranged these panels and edited the accompanying text to make clear that these demonstrate a property of the batch correction, rather than the underlying data.

- Perhaps an issue of personal taste but some of the numbers written on some of the figures look a little strange. E.g. fig. 1(d). Maybe worth writing something like "n=396" instead of "396" and aligning these consistently in the different quadrants?

We agree with the reviewer and we have applied these suggested changes.

In conclusion, I think this paper needs to be published. The findings are important, the approaches are generally solid, and I think it will save a lot of people a lot of headaches moving forward.

We thank again this reviewer for his/her positive and encouraging remarks and for endorsing the publication of our manuscript.

Reviewer #2 (Remarks to the Author):

Dempster and colleagues have performed a comprehensive evaluation of the agreement between the two largest CRISPR-Cas9 gene loss-of-function screenings in cancer cell lines. The findings derived from this evaluation represent an important reference for the scientific community. Many research groups plan to use this valuable data from the Cancer Dependency Map project. Thus, understanding how reproducible are read-out from this platform is fundamental for the scientific community.

The current status of the manuscript is very good and of high quality. The article is well written and well organized. The statistical tests performed to address the different

comparisons are adequate. The different levels of comparisons between the two data sets are comprehensive.

Major point:

The whole study is focused in showing the level of agreement between the two datasets. In contrast the source of disagreement is attributed mainly to technical variation (batch effect): reagents, screening quality and time point of read-out which was corrected it using Combat. However, it would be expected a certain level of disagreement due to biological variability as well. For instance, it would be expected that not all but key essential metabolic genes could disagree between the two datasets if the media culture used to grow the cells in each screening had different composition of essential metabolites. The authors showed the source of technical variation using PCA and visual inspection but a more careful inspection of the PC loadings should be performed to dissect if it also includes some biological variation. Although they performed a broad exploration of biological variation of those hits only found in a single dataset using GO terms, specific and interesting cases might be investigated. Thus, a detailed functional analysis on the PC loadings should be performed that would help to describe the type of biological variability if any.

In summary, authors should include an extended interpretation of the biological variability to complete their study that it would help researchers to understand the scope of this data for their projects.

We thank the reviewer for this constructive remark. To address this point, we have added a novel analysis encompassing a comparison of scores for asparagine synthetase (ASNS) genes notably more dependent in media lacking asparagine genes and identified by Lagziel et al. (BMC Biology 2019). Results from this analysis are reported in the new Figure 5b panel described in the following text now added to the section “Elucidating sources of disagreement between the two datasets” of our manuscript:

Many other experimental differences may also contribute to differences in reported response. For example, Lagziel et al. showed that many metabolic gene dependency profiles in Achilles are related to screening media, with e.g. asparagine synthetase (ASNS) notably more dependent in media lacking asparagine (Lagziel

et al. 2019). The Broad Institute used provider-recommended media for all Achilles screens, while the Sanger Institute adapted cells to either RPMI or a fifty-percent mix of DMEM and F12. While DMEM lacks asparagine, both RPMI and F12 contain it; thus, ASNS is expected to be a strong dependency only in Broad screens, and only in DMEM or other asparagine-deficient media. We confirmed this result (Fig. 5b). The difference between ASNS dependency in DMEM and either RPMI or DMEM:F12 in Broad screens is significant (Student's t-test $p = 1.52 \times 10^{-10}$, $N = 100$ and $p = 0.0173$, $N = 80$). In contrast, the difference between the RPMI and DMEM:F12 media conditions is not significant in either the Broad ($p = 0.961$, $N = 34$) or the Sanger ($p = 0.964$, $N = 147$). Although ASNS is the strongest example, it is likely that some of the differences in other metabolic genes between institutes are explained by media.

We have also conducted gene set enrichment analysis of genes sorted according to the loadings of the first two principal components using a comprehensive collection of 186 KEGG pathway gene sets from MsigDB. We found significant enrichment for genes involved in spliceosome and ribosome in the first principal component, indicating that screen quality likely explains some variability in the data (Supplementary Fig. 5a,b).

Minor points:

- *Figure 1 represents a good depiction of the problem and how they addressed it. However, the density of the scatter plot is saturated. It makes difficult to precisely visualize the agreement between the two datasets. Taking into account that the CRISPR screenings are performed in individual cell lines, one would expect to check this out in a single sample basis. Is there any cell line for which there is no agreement at all between the two datasets? I would suggest to add a supplementary table with Pearson's correlation coefficients per each cell line that was tested in both screenings. A supplementary boxplot with single-sample gene dependency correlations between the two screenings stratified by tissue*

might help the research community to figure out how “homogeneous” is the agreement across tissues and cell lines.

We agree with the reviewer that providing information about reproducibility would be useful and we have assembled a new data table where we report Pearson’s correlation coefficients per each cell line across the different considered gene-sets, i.e. all-genes, variable-genes, strongly-selective-dependency genes. This table is now integrated in Supplementary Table 5 and reports also the tissue of origin of the cell lines. Furthermore we have enriched Supplementary figure 3 with 2 additional panels. The first one shows a boxplot with single cell line gene dependency correlations between the two screens stratified by tissue, as suggested by the reviewer, and highlights that there are no tissues whose data is homogeneously more/less consistent than the rest of the tissues across the two studies. Furthermore, within each tissue, a strong determinant of screen consistency is still the inherent data quality of the screens performed in the individual studies, with poor agreement in individual lines generally explained by poor screen performance in at least one institute. All these results are now mentioned in the “Agreement of cell line dependency profiles” section, Supplementary Figure 3, and Supplementary Table 5, which includes also data quality assessment scores computed within the two studies, before and after batch correction. Particularly the following text is now included in our manuscript:

We finally examined whether the residual disagreement in corrected data might be related to screen quality and if there are tissues for which corresponding cell lines showed a consistently higher/lower consistency across the two studies. We assessed screen quality by computing true positive rates (TPRs) for recovering common essential genes in each cell line with a fixed 5% false discovery rate (FDR), determined from the distribution of nonessential genes in the cell line. We found that mean screen quality is a strong predictor of screen agreement for both the uncorrected and batch-corrected data sets (p-values 2.06×10^{-35} , 4.74×10^{-35} and adjusted R-squared 0.65, 0.64 for uncorrected and batch-corrected respectively; **Supplementary Fig. 3c**). In addition, we observed no differences in screen agreement when stratifying cell lines based on their tissue of origin (**Supplementary Fig. 3d**), with screen quality being highly correlated with screen agreement invariantly across tissues (**Supplementary Fig. 3e** and **Supplementary Table 5**).

- *In the “Agreement of gene dependency biomarkers” section, the authors performed a genome-wide t-test for significant differences in dependency scores between the two cell populations. It was difficult to understand how they defined the two cell populations for the statistical test. It seems that they stratified cell lines by the presence of a feature of interest (biomarker). Authors should explain better this stratification at the beginning of the section.*

We do apologise for the lack of clarity. We have reworded the text mentioned by the reviewer to better specify the origin of the features considered in the analysis and how we tested them against the genes scores for SSD genes. It now reads as follows:

Following a similar approach to that presented by the Cancer Cell Line Encyclopedia and Drug Sensitivity in Cancer consortia²¹, we performed a systematic test for molecular-feature/dependency associations on the two datasets. To this aim we considered a set of *Cancer Functional Events* consisting of 578 molecular features selected in Iorio *et al.*²⁷ based on their clinical relevance and encompassing mutations in high-confidence cancer driver genes, amplifications/deletions of chromosomal segments recurrently altered in cancer, hypermethylated gene promoters, microsatellite instability status, and the tissue of origin of the cell lines (**Supplementary Table 5**). We considered each of these features in turn and observed its status in the cell lines screened at both Sanger and Broad. Based on this, cell lines were split into two groups (respectively with negative/positive feature) and each of the SSD genes was *t*-tested for significant differences in gene scores across the obtained two groups of cell lines.

- *Fig 6A must have resize to be squared. Otherwise patterns are exaggerated along the rotated data from one principal component (i.e. PC1 in the plot).*

Figure 6A has been modified as suggested.

Reviewer #3 (Remarks to the Author):

The manuscript by Dempster et al., provides an analysis of the reproducibility of two recent large scale CRISPR-Cas9 functional genomics studies from the Sanger and Broad Institutes. These complimentary efforts represent an extraordinary resource for the community. It is therefore completely justified and important to understand the reproducibility of these efforts but also their differences.

The authors conclude that in general these CRISPR-Cas9 screening efforts are highly concordant and identify two contributing factors, library effect and assay length, to observed disagreement in gene dependency scores. While this is useful knowledge for the research community who will be mining these datasets and planning functional genomics studies, library batch effect observed that arise from factors such as different numbers gRNA per gene, variable gRNA knockout efficiency, and other experimental design have been previously recognized. However, the application of a previously described method (ComBAT) to correct for batch effect is certainly novel and a relatively simple way to integrate screens from various groups.

Interestingly, while the authors are focusing on identifying similarities between the two screening efforts, one could equally pay attention to their differences and factors that could explain the variations (once the screens are normalized). I suspect that this could be the focus of a follow up study by this group or others.

This final remark is in line with a comment from reviewer #2. We have addressed this point by performing a novel analysis encompassing a comparison of scores for metabolism genes whose effect of fitness upon inactivation is more dependent on the media employed in the two screens and an additional analysis focusing on the Principal Component loads, which are reflective of the variability of the gene scores. Results from these analyses are reported in the new Figure 5b panel and described in the section “Elucidating sources of disagreement between the two datasets” of our manuscript.

Nevertheless, we agree with the reviewer while he/she points out that these aspects definitely deserve a closer look and this might be the focus of follow-up studies.

Major Issues

1 As mentioned above, the novelty of this study could be questioned. Contributions of library composition and assay length to study-to-study variability have been previously described. Aside from a verification that these previous studies are more or less in agreement, when applied to large scale efforts. Given that these two large scale efforts will be a source of important information for the community moving forward, a study rigorously comparing the two efforts is however fully justified.

We thank the reviewer for their comments. We agree that the factors contributing to study-to-study variability have been previously described. Nevertheless, we believe that our study offers several important novel findings to the community:

1. A confirmation that these large CRISPR screens are in agreement, a result which is particularly important given the confusion surrounding compound screens referenced by Reviewer 1 in his introductory comment;
2. A comprehensive framework for comparing datasets which we hope will serve as a guide for future comparisons of large-scale experiments and prevent similar confusions in the future;
3. We were able to isolate the most significant experimental factors leading to disagreement between the screens,. Specifically, while it is known that the sgRNA library affects CRISPR screening results, we show here that it is indeed a key experimental parameter.

Taken together, we believe these results provide sufficient originality to our study.

2. Figure 2B does not appear to be cited in the manuscript text.

We thank the reviewer for pointing out this oversight, which we have corrected.

3. Throughout the paper the terms “gene score” “gene dependency score” and “dependency score” appear to be used interchangeably. Do these all refer to the same metric? If so please clarify, and if not select one description and use throughout to increase clarity.

The reviewer is correct that our language should be consistent. We have edited to use “gene score” throughout.

4. While differences in sgRNA efficiency is likely a key contributor to observed differences in screen results, I do not think the data shown in Figure 4 demonstrates this very well. Fig. 4B does a pretty good job of showing potentially problematic genes based on sgRNA correlation—however, later in the figure (panel D) a completely different set of genes is shown. Why not continue with the first set of genes? Furthermore, for Fig 3D, while the text states that “reagent efficiency likely explains some differences” no data matching this description is provided as the corresponding panel simply shows differences in gene scores. This section needs improved presentation of how reagent efficiency differences affect the scoring of example genes.

We believe that this confusion arises because Fig 4 illustrates two different points: in the first two panels, it shows the effect of reagent consistency on the agreement between datasets, while the following panels focus on reagent efficacy. This is a distinction we failed to discuss properly in the text. Although it is reasonable to assume that most cases of reagent inconsistency arise from differing reagent efficacy, we can only establish this for sgRNAs where some independent estimate shows different efficacies. To clarify this point we have extended and reworded the main text section describing the results shown in Fig 4.

Minor Issues

1. In caption for Fig. 1B it reads “The distributions scores...” should this read “The gene scores...” or “The distribution of gene scores...”

This has been corrected as suggested.

2. In Fig. 1D it is not entirely clear that the labels used in panel 1C are meant to carry over.

We have clarified this visually and in the caption text.

3. When analyzing cell line dependencies, a threshold of <-0.7 was employed. What is the rationale for this threshold? The raw number itself will have limited meaning to most readers so more information is warranted.

The reviewer is correct that this number may not be meaningful to readers. We have changed this section to use a threshold for each processing method such that the false discovery rate of gene scores falling below it is 0.05 (estimated using the distribution of nonessential gene scores vs all gene scores).

4. In Fig. 2D what does the "Au" in the axis labels refer to?

Au stands for Arbitrary units, we have now included this in the axis labels.

5. The description of the data shown in Fig. 3B is confusing and therefore it is difficult to understand what value it adds.

The plots in figure panel 3B complete in a quantitative way what shown in 3A. In fact they highlight that not only the two studies unveil consistent sets of biomarker of gene essentiality, this consistency is proportional to the statistical stringency used to define these biomarkers. This means that top significant associations found in one study are more likely to be detected in the other study with respect to weaker associations. We acknowledge that both figure legend and text describing this figure were quite convoluted and we have reworded both for the sake of clarity.

6. While MYCN is depicted in Fig 3C it is not described in the text while all others shown are.

Thank you for noticing this oversight. We have now added a description of the MYCN dependency to the text.

Examples of expected associations included increased dependency on ERBB2 in ERBB2-amplified cell lines, increased dependency on beta-catenin in APC mutant cell lines and increased dependency on MYCN in Peripheral Nervous System cell lines.

7. Fig 3D is said in the text to depict “significant associations between gene expression and dependency” which is actually quite difficult to discern from the figure panel. The panel for 3D instead shows a study to study comparison of these correlation scores.

We agree with the reviewer that this could be clearer. We have included additional description in the text and figure caption to highlight both the significant associations between gene expression and gene dependency as well as the agreement between associations identified in both data sets.

Systematic tests of each correlation identified significant associations between gene expression and dependency. Further, as with the genomic biomarkers, we found significant overlap between gene expression biomarkers identified in each dataset (Fisher’s exact test p-value below machine precision), and strong overall agreement between gene expression markers and SSD genes dependency across datasets (Pearson’s correlation 0.804) (Fig. 3d) .

(d) Comparison of results across the two studies of a systematic correlation test between gene expression and dependency of SSD genes across the two studies. The grey dashed lines indicate the thresholds of significant correlations at 5% false discovery rate identified for each study.

8. Caption for Fig 4D uses the acronym “MESE” but not explanation is provided.

We regret this oversight, and have added text explaining that MESE stands for Median (Azimuth) Estimated sgRNA Efficacy.

9. Is the difference in the mean score for late essentials between studies statistically significant?

We confirmed that the difference is statistically significant and provided the corresponding p value (2.57×10^{-78}) using a Gaussian fit of the difference of medians of randomly constructed gene sets in the main text.

REVIEWERS' COMMENTS:

Reviewer #1 (Remarks to the Author):

I am satisfied with the authors edits and their reply to the reviews.

Reviewer #2 (Remarks to the Author):

The authors have addressed very satisfactorily all my comments (and other reviewers') and they have considerably improved the manuscript including new analyses, text and figures that address all raised points. In my opinion, this publication is a valuable reference to understand how reproducible are read-out from Cancer Dependency Map project which is a valuable resource for the scientific community. I suggest the acceptance of the manuscript for publication in Nature Communications.

Reviewer #3 (Remarks to the Author):

The authors have now addressed all my concerns for the manuscript.

REVIEWERS' COMMENTS:

Reviewer #1 (Remarks to the Author):

I am satisfied with the authors edits and their reply to the reviews.

Reviewer #2 (Remarks to the Author):

The authors have addressed very satisfactorily all my comments (and other reviewers') and they have considerably improved the manuscript including new analyses, text and figures that address all raised points. In my opinion, this publication is a valuable reference to understand how reproducible are read-out from Cancer Dependency Map project which is a valuable resource for the scientific community. I suggest the acceptance of the manuscript for publication in Nature Communications.

Reviewer #3 (Remarks to the Author):

The authors have now addressed all my concerns for the manuscript.

We thank all the reviewers for their positive final comments.